# Perioperative changes in left ventricular systolic function following surgical revascularization

**Michael C. Downey**[1,2]*, **Matthew Hooks**[1,2], **Amy Gravely**[3], **Niyada Naksuk**[4], **Melissa Buelt-Gebhardt**[3], **Selma Carlson**[1,2], **Venkat Tholakanahalli**[1,2], **Selçuk Adabag**[1,2]

1 Division of Cardiology, Minneapolis VA Health Care System, Minneapolis, MN, United States of America, 2 Division of Cardiology, Department of Medicine, University of Minnesota, Minneapolis, MN, United States of America, 3 Research Service, Minneapolis VA Health Care System, Minneapolis, MN, United States of America, 4 Division of Cardiovascular Medicine, Department of Internal Medicine, University of Nebraska Medical Center, Omaha, NE, United States of America

* downe089@umn.edu

**Data Availability Statement:** The data underlying the results presented in the study are available from the STICH trial investigators and the United States Government by request referencing: "The

## Abstract

### Background

Nearly 1/3rd of patients undergoing coronary artery bypass graft surgery (CABG) have left ventricular systolic dysfunction. However, the extent, direction and implications of perioperative changes in left ventricular ejection fraction (LVEF) have not been well characterized in these patients.

### Methods

We studied the changes in LVEF among 549 patients with left ventricular systolic dysfunction (LVEF <50%) who underwent CABG as part of the Surgical Treatment for Ischemic Heart Failure (STICH) trial. Patients had pre- and post-CABG (4 month) LVEF assessments using identical cardiac imaging modality, interpreted at a core laboratory. An absolute change of >10% in LVEF was considered clinically significant.

### Results

Of the 549 patients (mean age 61.4±9.55 years, and 72 [13.1%] women), 145 (26.4%) had a >10% improvement in LVEF, 369 (67.2%) had no change and 35 (6.4%) had >10% worsening of LVEF following CABG. Patients with lower preoperative LVEF were more likely to experience an improvement after CABG (odds ratio 1.36; 95% CI 1.21–1.53; per 5% lower preoperative LVEF; p <0.001). Notably, incidence of postoperative improvement in LVEF was not influenced by presence, nor absence, of myocardial viability (25.5% vs. 28.3% respectively, p = 0.67). After adjusting for age, sex, baseline LVEF, and NYHA Class, a >10% improvement in LVEF after CABG was associated with a 57% lower risk of all-cause mortality (HR: 0.43, 95% CI: 0.26–0.71).

Comparison of Surgical and Medical Treatment for Congestive Heart Failure and Coronary Artery Disease (STICH)" found at ClinicalTrials.gov and with identifier: NCT00023595.

**Funding:** SA received an education grant from Medtronic (https://www.medtronic.com/us-en/index.html) The funders had no role in study design, data collection and analysis, decision to publish, or preparation of the manuscript.

**Competing interests:** The authors have declared that no competing interests exist.

## Conclusions

Among patients with ischemic cardiomyopathy undergoing CABG, 26.4% had >10% improvement in LVEF. An improvement in LVEF was more likely in patients with lower pre-operative LVEF and was associated with improved long-term survival.

## Introduction

Coronary artery bypass graft (CABG) surgery improves the long-term survival of patients with left main and/or multi-vessel coronary artery disease with reduced left ventricular (LV) systolic function [1]. Nearly 1/3rd of patients undergoing CABG have LV systolic dysfunction with ejection fraction <50% [2]. Although there is reason to expect that reduction of myocardial ischemia and recovery of hibernating myocardium through coronary revascularization would result in improvement of LV systolic function, there is relatively little data to support this assertion [3]. Prior single-center, retrospective studies in this area were limited by patient selection bias and imaging studies that were not performed systematically at pre-determined time points after CABG [4–6]. Since left ventricular ejection fraction (LVEF) is an important clinical variable guiding therapeutic decisions, and offering prognostic information, it is important to characterize the extent, direction and implications of LVEF changes following CABG [7–9]. In this study, we assessed perioperative changes in LVEF among patients randomized to CABG in the Surgical Treatment for Ischemic Heart Failure (STICH) trial [10–12].

## Materials and methods

### Patient population

The STICH trial had 2 hypotheses and included 2,136 patients with LV systolic dysfunction, and coronary artery disease amendable to CABG [13]. The 1st hypothesis included 1,212 patients randomized to CABG plus guideline-directed medical therapy versus medical therapy alone. At 10-year follow-up, patients assigned to CABG had significantly lower rates of all-cause mortality, cardiovascular mortality, and hospitalizations compared to those assigned to medical therapy [12]. The 2nd hypothesis included 1,000 patients randomized to either CABG with surgical ventricular restoration (SVR) or CABG alone. The results showed that addition of SVR to CABG made no difference in outcomes [10].

From the 1,000 patients enrolled in the trial to test the 2nd hypothesis (CABG with SVR vs. CABG alone) 770 (77%) had an LVEF assessment at baseline and 4 months postoperatively, interpreted at a STICH core laboratory [10–12]. All patients had evidence of systolic dysfunction (LVEF <50%) before CABG [14]. We excluded patients who had suboptimal image quality (*n* = 181). Additionally, any patients where there was a mismatch between the pre- and postoperative imaging modality were excluded (*n* = 40). The final cohort for this *post-hoc* analysis included 549 patients who underwent CABG (+/- SVR), had pre- and postoperative LVEF assessment via identical imaging modalities with good-excellent image quality, evaluated at a STICH core laboratory.

### Imaging assessment of LVEF

In the STICH trial, LVEF was determined by echocardiography (echo), cardiac magnetic nuclear resonance imaging (CMR), or radionucleotide imaging (RN), as previously described

[14]. Interpretation of the acquired images was performed at central core laboratories. The readers were blinded to the patients' clinical information and treatment assignment. The preoperative LVEF assessment was required within 3 months of trial entry.

### Definition of LVEF change

Change in LVEF (ΔLVEF) was defined as: *Postoperative LVEF−Preoperative LVEF*. LVEF assessment via echo has been reported to have a test-retest reliability of ±5%, predisposing analyses conducted at lower thresholds to type I errors [15]. As such, in this analysis we defined clinically significant ΔLVEF as >10%.

### Myocardial viability

Although myocardial viability testing was initially a requirement for all patients, the STICH trial protocol was subsequently revised to make it optional as it proved to be an impediment to patient enrollment. Viability testing was done using either single-photon emission computed tomography or dobutamine stress echo, depending on the availability of the technique and expertise at recruiting centers. The interpretation and analysis of the viability studies were done at core laboratories as previously described [3].

### Statistical analysis

Categorical variables are reported as frequency (%) and continuous variables as mean +/- standard deviation. The patients were classified as "Improved LVEF," "Decreased LVEF," and "Unchanged LVEF" based on ΔLVEF >10% (>5% in sensitivity analysis). Relationships between variables of interest and categories of ΔLVEF were tested with chi-square tests for categorical variables and ANOVA for continuous variables. We utilized logistic regression analysis to examine the predictors of EF improvement. All variables that had a p-value $\leq 0.10$ in univariable analysis were entered into a multivariable logistic regression model (S1 Table). Utilizing backwards elimination, and a more restrictive p-value of $\leq 0.05$, we reached the final multivariable model [16]. Kaplan-Meier survival analysis was performed to illustrate all-cause mortality in relation to perioperative LVEF improvement >10%. The survival curves were compared using log-rank test. Multivariable Cox regression analysis was used to assess the hazard ratio (HR) of all-cause mortality associated with perioperative LVEF improvement. Survival analysis was adjusted for all covariates which were significantly different between patients with vs. without improved LVEF. Analyses were performed using SAS® 9.4. All analyses were 2-sided and a p-value < 0.05 was taken as significant.

### Ethics approval

All data used in this study has been de-identified according to the Health Insurance Portability and Accountability Act of 1996 (HIPAA) 164.514 Privacy Rule. The study was performed in accordance with the ethical standards as laid down in the 1964 Declaration of Helsinki and its later amendments. As this study is analyzing de-identified, publicly-available data, it is exempt from Institutional Review Board approval; but the Research and Development Committee of the Minneapolis Veterans Affairs Medical Center approved this analysis.

### Consent to participate and publish

Informed consent and consent to publish were obtained from all individual participants as a required component of enrollment in the STICH trial [13].

## Results

The baseline characteristics of the 549 study patients who underwent CABG (+/- SVR) are shown in Table 1. Mean patient age (±SD) was 61.4 (±9.6) years, and 72 (13.1%) were women. A total of 258 (47.0%) patients underwent concurrent CABG with SVR. LVEF assessment was made by echo in 273 (49.7%), CMR in 191 (34.8%), and RN in 85 (15.5%) of the patients (Table 1).

### Perioperative changes in LVEF

Following CABG, 145 (26.4%) patients had improvement in LVEF >10%, 369 (67.2%) had no change, and 35 (6.4%) had decrease in LVEF >10% (Fig 1). For the patients who experienced LVEF improvement, the mean LVEF increased from 25.1% (±9.1%) to 42.8% (±10.9%). Among those with worsening of LVEF, the mean LVEF decreased from 36.5% (±7.5%) to 20.5% (±6.5%).

**Table 1. Baseline and intraoperative characteristics of study patients by change in LVEF.**

| Variable | | All Patients | Improved LVEF | Unchanged LVEF | Decreased LVEF | P-Value |
|---|---|---|---|---|---|---|
| | | N = 549 | N = 145 | N = 369 | N = 35 | |
| Age, years | | 61.43 ± 9.55 | 62.2 ± 9.6 | 61.23 ± 9.54 | 60.38 ± 9.51 | 0.47 |
| Women, n (%) | | 72 (13.1) | 21 (14.5) | 48 (13.0) | 3 (8.6) | 0.65 |
| Caucasian, n (%) | | 506 (92.2) | 134 (92.4) | 339 (91.9) | 33 (94.3) | 0.87 |
| Imaging Modality | Echo, n (%) | 273 (49.7) | 58 (40.0) | 196 (53.1) | 19 (54.3) | 0.005 |
| | CMR, n (%) | 191 (34.8) | 60 (41.4) | 115 (31.2) | 16 (45.7) | |
| | RN, n (%) | 85 (15.5) | 27 (18.6) | 58 (15.7) | 0 (0.0) | |
| Preoperative EF | | 28.59 ± 9.11 | 25.08 ± 9.05 | 29.22 ± 8.67 | 36.52 ± 7.51 | <0.001 |
| History of MI, n (%) | | 472 (86.0) | 123 (84.8) | 318 (86.2) | 31 (88.6) | 0.83 |
| History of diabetes, n (%) | | 171 (31.2) | 44 (30.3) | 117 (31.7) | 10 (28.6) | 0.90 |
| History of hypertension, n (%) | | 294 (53.6) | 72 (49.7) | 209 (56.6) | 13 (37.1) | 0.05 |
| Body mass index | | 27.15 ± 4.2 | 27.2 ± 4.49 | 27.17 ± 4.13 | 26.81 ± 3.72 | 0.88 |
| NYHA class III/IV, n (%) | | 235 (42.8) | 72 (49.7) | 151 (40.9) | 12 (34.3) | 0.11 |
| Creatinine | | 1.12 ± 0.37 | 1.13 ± 0.41 | 1.11 ± 0.32 | 1.21 ± 0.6 | 0.34 |
| Beta-blocker, n (%) | | 485 (88.3) | 127 (87.6) | 328 (88.9) | 30 (85.7) | 0.81 |
| ACE-I or ARB, n (%) | | 487 (88.7) | 125 (86.2) | 333 (90.2) | 29 (82.9) | 0.23 |
| Aspirin, n (%) | | 435 (79.2) | 113 (77.9) | 296 (80.2) | 26 (74.3) | 0.64 |
| Clopidogrel, n (%) | | 40 (7.3) | 13 (9.0) | 25 (6.8) | 2 (5.7) | 0.65 |
| Digoxin, n (%) | | 88 (16.0) | 17 (11.7) | 66 (17.9) | 5 (14.3) | 0.22 |
| Diuretic loop, n (%) | | 319 (58.1) | 87 (60.0) | 208 (56.4) | 24 (68.6) | 0.33 |
| K sparing diuretic, n (%) | | 211 (38.4) | 56 (38.6) | 144 (39.0) | 11 (31.4) | 0.68 |
| Nitrate, n (%) | | 316 (57.6) | 82 (56.6) | 213 (57.7) | 21 (60.0) | 0.93 |
| Pulse | | 71.96 ± 12.24 | 73.3 ± 14.09 | 71.3 ± 11.41 | 73.26 ± 12.33 | 0.20 |
| Systolic BP | | 120.23 ± 17.08 | 118.95 ± 17.31 | 120.94 ± 16.89 | 118.03 ± 18.07 | 0.36 |
| No. of distal anastomoses | | 3.12 ± 1.06 | 3.12 ± 0.96 | 3.13 ± 1.08 | 3.06 ± 1.24 | 0.93 |
| No. of diseased vessels | | 2.17 ± 0.78 | 2.19 ± 0.83 | 2.17 ± 0.75 | 2.17 ± 0.82 | 0.96 |
| Total bypass time, min | | 116.44 ± 45.98 | 120.17 ± 45.06 | 115.9 ± 46.87 | 106.36 ± 39.44 | 0.28 |
| Aortic cross-clamp time, min | | 75.84 ± 32.14 | 77.45 ± 30.99 | 75.74 ± 32.69 | 70.18 ± 31.53 | 0.51 |
| CABG + SVR, n (%) | | 258 (47.0) | 84 (57.9) | 164 (44.4) | 10 (28.6) | 0.002 |

Abbreviations: ACE-I = angiotensin converting enzyme inhibitor, ARB = angiotensin receptor blocker, BP = blood pressure, CABG = coronary artery bypass graft, CMR = cardiac magnetic resonance, Echo = transthoracic echocardiogram, K = potassium, LVEF = left ventricular ejection fraction, MI = myocardial infarction, No. = number, NYHA = New York Heart Association, RN = Radionuclide, SVR = surgical ventricular restoration

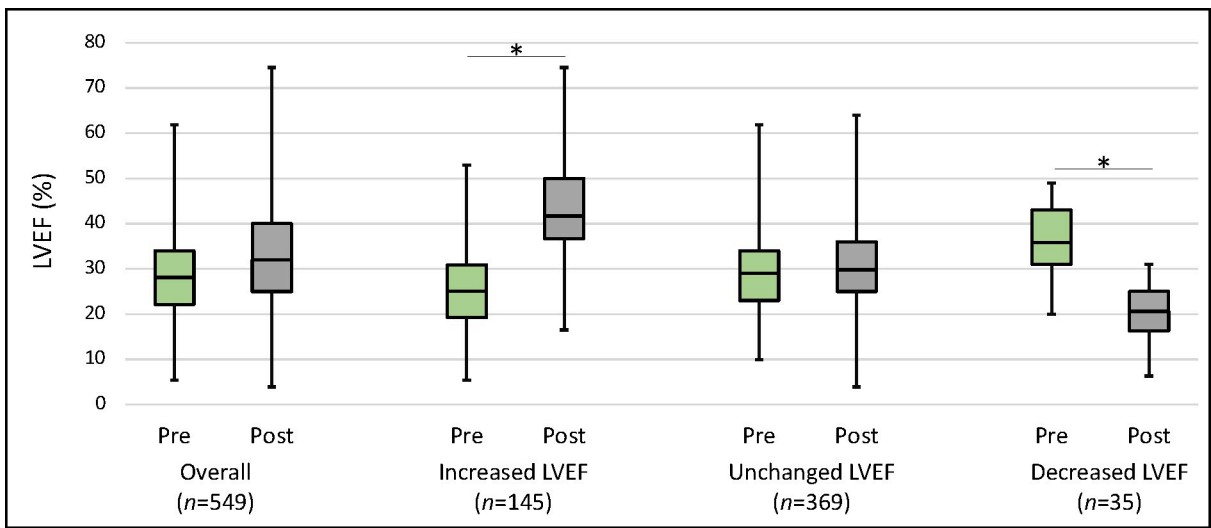

**Fig 1. Distribution of pre- and postoperative LVEF stratified by a ΔLVEF >10%.** Box borders represent 1st and 3rd quartiles with bisecting line representing the median. Whiskers demarcate minimum and maximum values. *denotes *p*<0.001. *Abbreviations*: LVEF = left ventricular ejection fraction.

Notably, there was an inverse association between preoperative LVEF and the likelihood of LVEF improvement >10% (Fig 2). Of the patients with preoperative LVEF ≤20%, 42.2% (n = 43) had >10% LVEF improvement. As preoperative LVEF increased, there was a stepwise decline in the incidence of LVEF improvement (Fig 2). The converse occurred with LVEF worsening >10%. As the preoperative LVEF increased, there was a stepwise increase in the

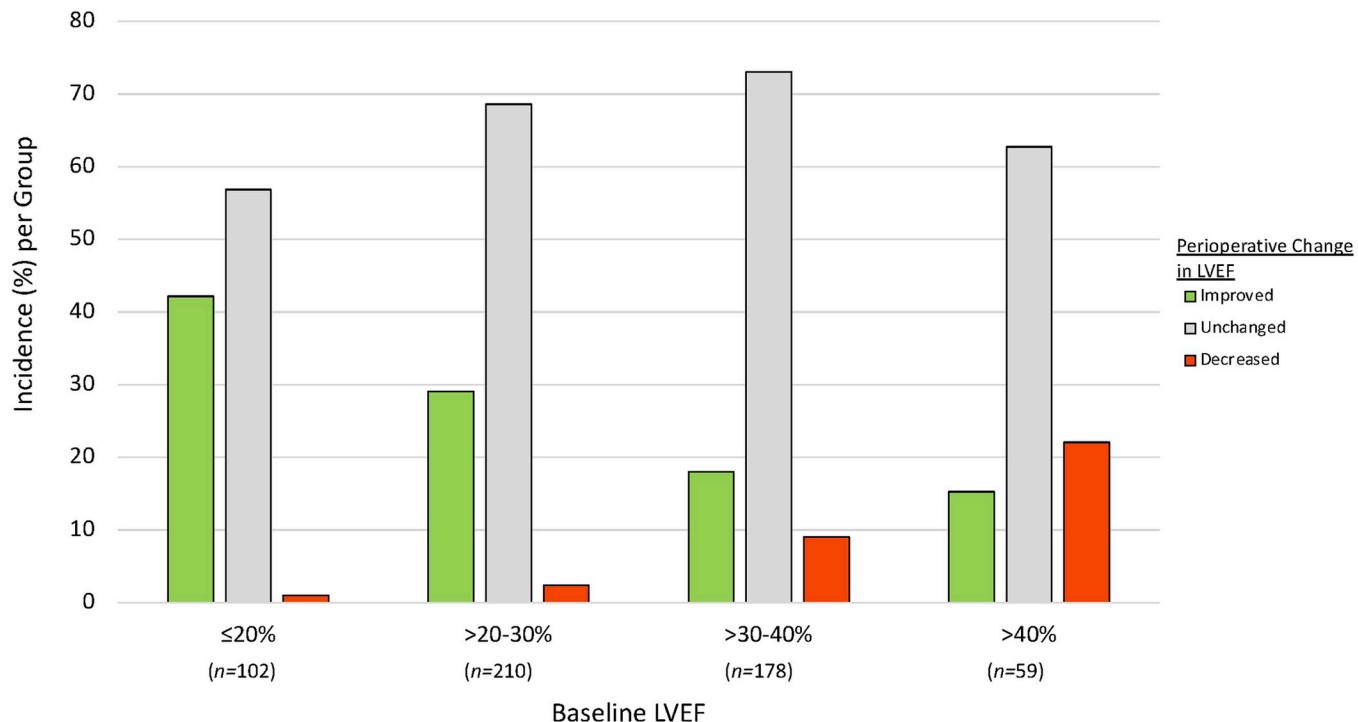

**Fig 2. Perioperative change in LVEF by baseline LVEF.** *Abbreviations*: LVEF = left ventricular ejection fraction.

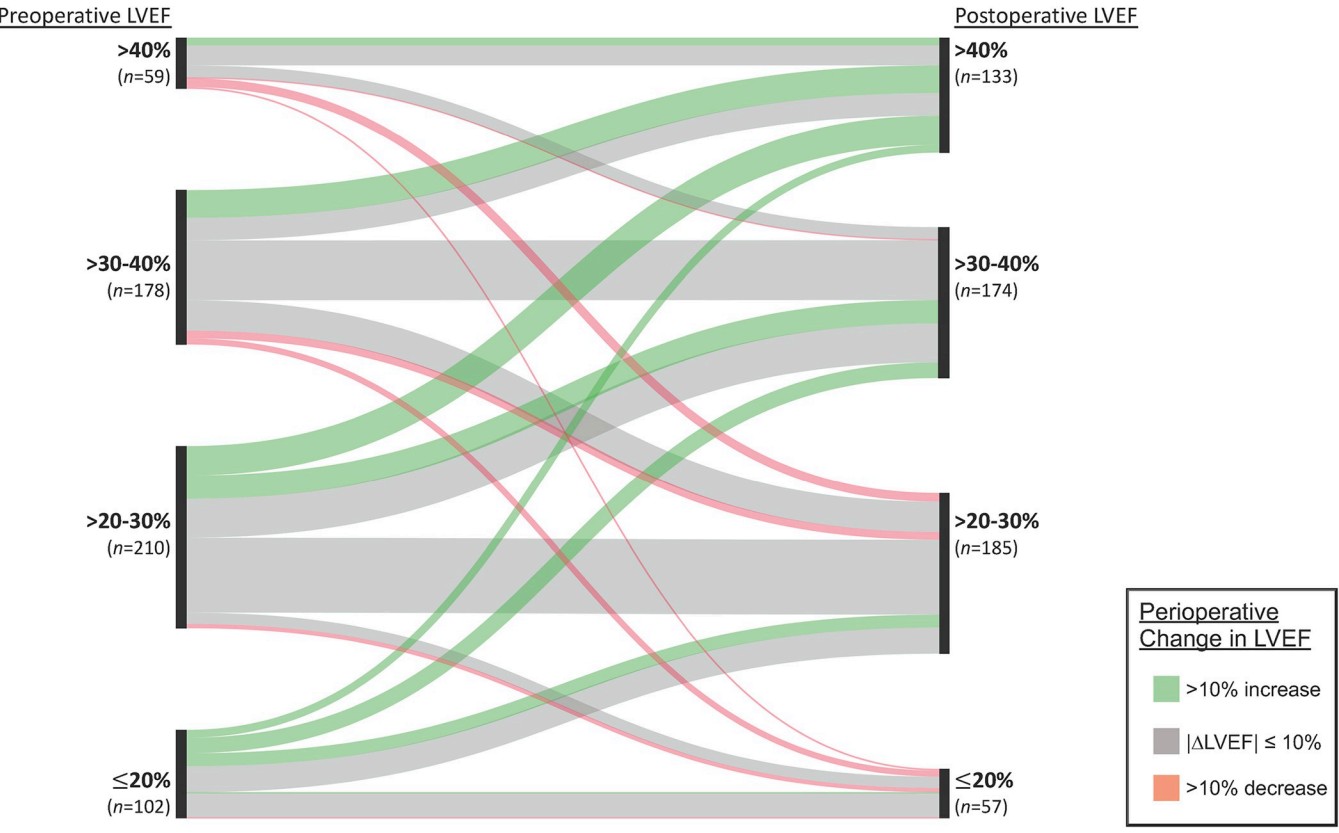

**Fig 3. Sankey flow diagram of change in LVEF pre- to post-CABG.** Pre-CABG LVEF by binned percentage ranges on the left axis, and Post-CABG LVEF comparably on the right axis. Flow follows left to right. Paratheses on axis represent total number of patients within each bin. Color of flow represents subgroup's perioperative change in LVEF. *Abbreviations*: CABG = coronary artery bypass graft, LVEF = left ventricular ejection fraction.

incidence of LVEF worsening (Fig 2). A flow diagram of pre- and postoperative LVEF are shown in Fig 3.

In multivariate logistic regression analysis, preoperative LVEF and SVR were independent predictors of LVEF improvement >10% following CABG. The odds of LVEF improvement were 1.36 times higher (95% CI 1.21–1.53; p <0.001) per 5% decrease in preoperative LVEF (Table 2).

### Effect of myocardial viability on LVEF improvement

A total of 217 (39.5%) patients had preoperative myocardial viability test. Of these, 157 (72.4%) showed myocardial viability and 60 (27.7%) did not. Improvement in LVEF occurred

**Table 2. Multivariate logistic regression for independent predictors of >10% increase in LVEF.**

| Variable | OR | 95% CI | P-value |
|---|---|---|---|
| Preoperative LVEF | 1.36* | (1.21–1.53) | <0.001 |
| SVR | 1.76 | (1.18–2.61) | 0.005 |

*Represents the odds of >10% perioperative increase in LVEF improvement per each decrease of 5% percentage points change in preoperative LVEF. Abbreviations: CABG = coronary artery bypass graft, LVEF = left ventricular ejection fraction, SVR = surgical ventricular restoration

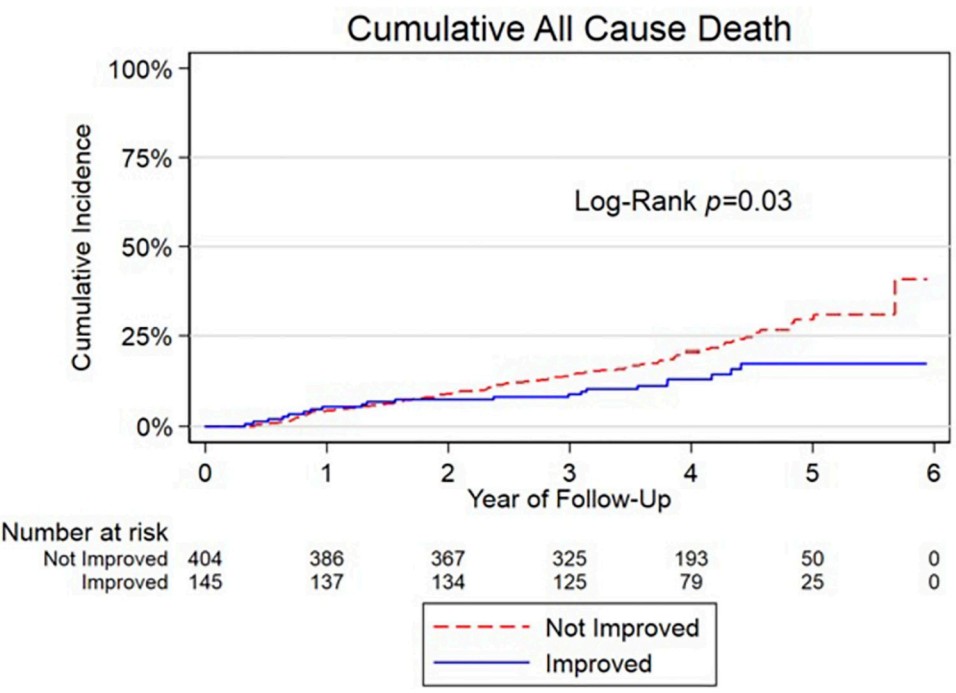

**Fig 4. Long-term survival in relation to perioperative LVEF improvement.**

in 40 (25.5%) of patients with myocardial viability versus 17 (28.3%) patients without myocardial viability (p = 0.67).

## Perioperative change in LVEF and survival

Over a mean 3.7 (±1.2) years of follow-up, 21/145 (14.5%) vs. 93/404 (23.0%) of patients with or without a ΔLVEF >10% died, respectively. This translated to a significantly lower risk of all-cause mortality in patients with a perioperative LVEF improvement >10% compared to those with unchanged or decreased LVEF (p = 0.027) (Fig 4). After adjusting for age, sex, baseline LVEF, and NYHA Class, perioperative LVEF improvement >10% was associated with a 57% lower risk of all-cause mortality (HR: 0.43, 95% CI: 0.26–0.71) compared to those with unchanged or decreased LVEF.

In a competing risk analysis, perioperative LVEF improvement >10% was not associated with the risk of heart failure death (HR: 0.78, 95% CI 0.35–1.71; p = 0.53) or sudden cardiac death (SCD) (HR: 0.62, 95% CI 0.26–1.50; p = 0.29), though the statistical power of these analyses was low (36 and 33 total deaths, respectively).

## Sensitivity analysis

A sensitivity analysis was performed wherein ΔLVEF was redefined as >5%. In this analysis, 240 (43.7%) patients experienced improvement in LVEF, 220 (40.1%) had no change and 89 (16.2%), had worsening of LVEF. The results were otherwise similar to the main analysis.

## Discussion

In this *post-hoc* analysis of the STICH trial 326% of the patients with ischemic cardiomyopathy randomized to CABG had a perioperative increase in LVEF >10%, 67% had no change and 6% had a decline in LVEF >10%. The independent predictors of LVEF improvement were

preoperative LVEF (inverse association) and concurrent SVR. However, myocardial viability was not a factor. Notably, perioperative LVEF improvement >10% conferred a significant mortality benefit relative to those in whom LVEF remained unchanged or worsened.

As postoperative LVEF assessment is not routinely performed after CABG, prior investigations in this area have largely been limited to single center, retrospective studies with modest sample size [17]. Koene *et al.* [4], evaluated 375 patients wherein half of the patients had a preoperative LVEF <50%. Utilizing a ΔLVEF >5% cut off, 24% of patients had improvement in LVEF (vs. > 40% of the patients in the current analysis). Patients with a preoperative LVEF <50% were more likely to have LVEF improvement; while those with preoperative LVEF >50% were more likely to experience a decline in LVEF [4]. Papestiev *et al.*, prospectively evaluated 47 patients (27 with preoperative LVEF >50%) and found that ΔLVEF >5% occurred in 53.2% of patients. LVEF improvement of >5% was significantly more likely if LVEF <50%, and preoperative LVEF was inversely associated with perioperative LVEF improvement [6]. Similarly, in two cohort studies limited to preoperative LVEF <35%, CABG was associated with increases in LVEF [18, 19]. Finally, within a cohort with LVEF <50% (mean LVEF = 32%), Cornel *et al.* observed a ΔLVEF >5% in 19% of patients at 3 months, which increased to 31% at twelve months [20]. Cumulatively, these prior works have shown LVEF improvement in patients who had preoperative LV systolic dysfunction but a higher risk of LVEF decline in individuals with a preoperatively normal LVEF [4–6, 18–20].

In this analysis, we found that a perioperative LVEF improvement >10% afforded a 57% reduction in all-cause mortality after CABG. These results are similar to a 39% reduction in mortality following ΔLVEF ≥10% over a two year interval found in a contemporary analysis by Perry *et al.* [21]. Interestingly, within the hypothesis 1 cohort, while both CABG and ΔLVEF >10% reduced long term mortality, these effects were independent of the other [21]; suggesting that the mortality benefit afforded by CABG was not directly associated with improvement in LVEF and vice versa. Important differences between the work by Perry *et al.* and the current study include the timing of postoperative LVEF assessment (4 months vs. 24 months) as well as systematic pathophysiological differences between patients included in hypothesis 1 analysis (Perry *et al.*) vs. hypothesis 2 (current analysis) analysis of the STICH trial. All patients within the hypothesis 2 arm of STICH had evidence of dominant anterior wall akinesia or dyskinesia [a requirement to be eligible for SVR], versus only 12% of patients in the hypothesis 1 arm [22].

While SVR was identified as an independent predictor of ΔLVEF >10% and reduced mortality risk, the principle analysis of the hypothesis 2 data from the STICH trial did not find a mortality benefit with SVR beyond that provided by CABG alone [10]. This absence of direct effect of SVR on mortality suggests that while SVR may increase the *odds* of ΔLVEF >10%, it is only by *achieving* a ΔLVEF >10% the mortality benefit is realized; SVR itself is neither necessary nor sufficient to improve mortality in the absence of LVEF improvement.

In this study, we did not observe a protective effect of LVEF improvement on the incidence of SCD, but the analysis lacked statistical power. Previous studies have shown that improvement in LVEF is associated with reduced risk of SCD [23–25]. Our results build on a previous analysis of SCD from the STICH trial by Rao *et al.* [26]. Analyzing all 1,411 patients who underwent CABG across the STICH trial, Rao *et al.* observed 113 occurrences of SCD over 5 years for an 8.5% 5-year cumulative incidence of SCD [26]. Notably, SCD risk was greatest in the postoperative window from 30–90 days. Additionally, while lower preoperative LVEF predicted increased risk of perioperative SCD, increased preoperative end-systolic volume index and B-type natriuretic peptide were the most robust independent predictors of SCD risk [26]. Given the highest risk of SCD was within the first 3 months of CABG, post-CABG LVEF assessments at 4 months, as in current analysis, would fail to capture the LVEF change within

most individuals experiencing SCD [27, 28]. Future studies with more proximal postoperative LVEF assessments may better assess how ΔLVEF affects SCD risk.

## Limitations

This post-hoc analysis of a large randomized clinical trial data has several limitations. First, Caucasian men comprised >80% of the study patients. Caution is recommended when extending these results to women and minorities. Second, postoperative imaging captured an incomplete subset (77%) of the patients enrolled in the STICH trial [11]. Part of this deficit is attributable to mortality within 4 months of CABG, prior to assessment of postoperative LVEF [14, 26]. However, since 4 months was set in the study protocol prior to randomization, a systematic bias in patient selection for imaging studies is unlikely. Third, even though a majority of LVEF assessment was by echo, two other imaging modalities were also used, creating the possibility of inter-modality differences in LVEF. It has been previously demonstrated that while substantive variation can occur *between* modalities, no one modality consistently over or underestimates LVEF [29]. To protect against inter-modality differences this analysis only included data from patients with identical pre- and postoperative imaging modalities. Fourth, longitudinal interval data of LVEF was not available. However, a recent analysis of data from the Sudden Cardiac Death in Heart Failure Trial (SCD-HeFT) demonstrated that LVEF may oscillate over time in a portion of patients with cardiomyopathy. Those with initial increase in LVEF may then experience a subsequent decrease in LVEF, and vice versa [30]. Future studies investigating if a postoperative ΔLVEF >10% is sustained would be interesting. Finally, guideline-directed medical therapy has evolved since the era of the STICH trial. With the discovery of angiotensin receptor-neprilysin inhibitors [31] and sodium-glucose cotransporter 2 inhibitors [32] further increases in the proportion of patients experiencing LVEF and improved mortality are expected.

## Conclusions

In conclusion, approximately 25% of the patients with ischemic cardiomyopathy undergoing CABG experienced a >10% perioperative improvement in LVEF. The likelihood of LVEF improvement was inversely proportional to the preoperative LVEF. Improvement in LVEF was not influenced by the presence nor absence of myocardial viability. Improvement in LVEF was associated with better long-term survival. These results further build on the understanding of CABG associated perioperative change in LVEF by identifying the inverse relationship between improvement in LVEF and preoperative LVEF, which can further inform patient-physician decision marking around CABG.

## Supporting information

**S1 Table. Odds ratios of battery of potential predictors for LVEF improvement.**
(DOCX)

**S2 Table. Pre- and post-CABG LVEF by imaging modality.**
(DOCX)

## Author Contributions

**Conceptualization:** Michael C. Downey, Matthew Hooks, Amy Gravely, Niyada Naksuk, Selma Carlson, Venkat Tholakanahalli, Selçuk Adabag.

**Data curation:** Melissa Buelt-Gebhardt.

**Funding acquisition:** Selçuk Adabag.

**Investigation:** Michael C. Downey, Amy Gravely, Selçuk Adabag.

**Methodology:** Michael C. Downey, Matthew Hooks, Amy Gravely, Niyada Naksuk, Selma Carlson, Selçuk Adabag.

**Project administration:** Selçuk Adabag.

**Supervision:** Selçuk Adabag.

**Validation:** Matthew Hooks, Amy Gravely, Niyada Naksuk, Selma Carlson, Selçuk Adabag.

**Visualization:** Michael C. Downey, Amy Gravely, Selçuk Adabag.

**Writing – original draft:** Michael C. Downey, Matthew Hooks, Amy Gravely, Niyada Naksuk, Melissa Buelt-Gebhardt, Selma Carlson, Venkat Tholakanahalli, Selçuk Adabag.

**Writing – review & editing:** Michael C. Downey, Matthew Hooks, Amy Gravely, Niyada Naksuk, Melissa Buelt-Gebhardt, Selma Carlson, Venkat Tholakanahalli, Selçuk Adabag.

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
