## [Decision Letter · Decision Letter 0]

22 Aug 2022

PONE-D-22-19400Perioperative changes in left ventricular systolic function following surgical revascularizationPLOS ONE

Dear Dr. Downey,

Thank you for submitting your manuscript to PLOS ONE. After careful consideration, we feel that it has merit but does not fully meet PLOS ONE’s publication criteria as it currently stands. Therefore, we invite you to submit a revised version of the manuscript that addresses the points raised during the review process.

We look forward to receiving your revised manuscript.

Kind regards,

Yoshihiro Fukumoto

Academic Editor

PLOS ONE

Journal Requirements:

Reviewers' comments:

Reviewer's Responses to Questions

**Comments to the Author**

1. Is the manuscript technically sound, and do the data support the conclusions?

Reviewer #1: Yes

Reviewer #2: Partly

Reviewer #3: Yes

2. Has the statistical analysis been performed appropriately and rigorously? 

Reviewer #1: Yes

Reviewer #2: No

Reviewer #3: Yes

3. Have the authors made all data underlying the findings in their manuscript fully available?

Reviewer #1: Yes

Reviewer #2: Yes

Reviewer #3: Yes

4. Is the manuscript presented in an intelligible fashion and written in standard English?

Reviewer #1: Yes

Reviewer #2: Yes

Reviewer #3: Yes

5. Review Comments to the Author

Reviewer #1: The authors examined changes in LVEF in 549 patients with left ventricular systolic dysfunction (LVEF <50%) who underwent CABG as part of the Surgical Treatment of Ischemic Heart Failure (STICH) trial.

In conclusion, approximately 25% of patients with ischemic cardiomyopathy undergoing CABG experienced a perioperative improvement in LVEF of 10% or more; the likelihood of LVEF improvement was inversely proportional to preoperative LVEF; LVEF improvement was unaffected by the presence of myocardial viability; LVEF improvement was associated with improved long-term survival and improvement was associated with improved long-term survival. These results further our understanding of perioperative LVEF changes related to CABG by demonstrating an inverse relationship between LVEF improvement and preoperative LVEF.

The presentation is difficult to read because of the inclusion of Tables and Figure legends in the Results, which is different from the usual presentation. Other than the above, the logic is consistent and the discussion is adequate.

Reviewer #2: PONE-D-22-19400: statistical review

SUMMARY. This is a retrospective study of the changes in left ventricular ejection fraction (LVEF) among patients with systolic dysfunction who underwent coronary artery bypass graft surgery (CABG). The core statistical analysis relies on logistic regression (to examine predictors of ejection fraction improvements) and survival analysis (to estimate the association between perioperative LVEF improvement and all-cause mortality). Methods are correct and results seem sound. However, the presentation of the material is a bit too synthetic and further details are required: see the issues below.

MAJOR ISSUES

1. Table 2 display the results of a multivariable logistic regression model, obtained by a backward selection of the predictors. However, there is no information about (1) the initial battery of the predictors and (2) the final battery of the selected predictors. This information must be added. Specifically, Table 2 shouldn't only include the ORs of Preoperative LVEF and SVR but also the ORs of the remaining selected predictors included in the final model.

2. The analysis of perioperative change in LVEF and survival has been adjusted with respect to age and gender. What about all the other predictors displayed in Table 1? I'd welcome a Cox regression model where the predictors were selected by backward elimination, similarly to the approach pursued in the logistic regression analysis. A table with the results of the final Cox model should be then added and discussed. An additional specific point: Fig. 4 displays cumulative incidence curves. Why didn't the authors provide traditional survival curves?

3. The competing risks analysis doesn't seem to be adjusted for the available predictors. Why? I'd welcome an analysis that adjusts for the available predictors, consistently to what has been done in the rest of the paper.

Reviewer #3: This very interesting study evaluates the impact of myocardial revascularization on left ventricular function. Even though, a few questions must be answered before acceptance.

1) This study is a post-hoc analysis of the STICH trial (2º hypothesis). Although it is implied in the manuscript that this research is a post-hoc analysis of a portion of the STICH trial, this is not clearly stated in the method section of the manuscript.

2) Inclusion criteria are not clearly stated in the manuscript.

3) Another question that must be answered is why the patients participating in the first hypothesis of were not included.

4) The study used different methods to evaluate LVEF before and after the procedure. The variations observed in LVEF were comparable among the different methods?

5) There is no mention in the manuscript how the authors defined the sample size or reason for not having it defined.

6) SVR did not promote improvement in the original stich trial. How come that it was an independent predictor for > 10 % increase in LVEF? Selection Bias?

6. PLOS authors have the option to publish the peer review history of their article (what does this mean?). If published, this will include your full peer review and any attached files.

Reviewer #1: No

Reviewer #2: No

Reviewer #3: No

---

## [Author Response · Author response to Decision Letter 0]

23 Sep 2022

We thank the reviewers for their time and insights to improve our work. Below is an itemized response to their comments, including the modifications made to the manuscript in turn.

Reviewer #1: 

The authors examined changes in LVEF in 549 patients with left ventricular systolic dysfunction (LVEF <50%) who underwent CABG as part of the Surgical Treatment of Ischemic Heart Failure (STICH) trial.

In conclusion, approximately 25% of patients with ischemic cardiomyopathy undergoing CABG experienced a perioperative improvement in LVEF of 10% or more; the likelihood of LVEF improvement was inversely proportional to preoperative LVEF; LVEF improvement was unaffected by the presence of myocardial viability; LVEF improvement was associated with improved long-term survival and improvement was associated with improved long-term survival. These results further our understanding of perioperative LVEF changes related to CABG by demonstrating an inverse relationship between LVEF improvement and preoperative LVEF.

The presentation is difficult to read because of the inclusion of Tables and Figure legends in the Results, which is different from the usual presentation. Other than the above, the logic is consistent and the discussion is adequate.

Response: We thank the reviewer and agree that inclusion of tables and figure legends is different than usual. We included the tables and figure legends directly in the text per the PLOSone guidelines. (https://journals.plos.org/plosone/s/file?id=wjVg/PLOSOne_formatting_sample_main_body.pdf)

Reviewer #2:

SUMMARY. This is a retrospective study of the changes in left ventricular ejection fraction (LVEF) among patients with systolic dysfunction who underwent coronary artery bypass graft surgery (CABG). The core statistical analysis relies on logistic regression (to examine predictors of ejection fraction improvements) and survival analysis (to estimate the association between perioperative LVEF improvement and all-cause mortality). Methods are correct and results seem sound. However, the presentation of the material is a bit too synthetic and further details are required: see the issues below.

MAJOR ISSUES

1. Table 2 display the results of a multivariable logistic regression model, obtained by a backward selection of the predictors. However, there is no information about (1) the initial battery of the predictors and (2) the final battery of the selected predictors. This information must be added. Specifically, Table 2 shouldn't only include the ORs of Preoperative LVEF and SVR but also the ORs of the remaining selected predictors included in the final model.

Response: All variables in Table 1 (and 20 additional variables) were assessed with univariable logistic regression analysis. These variables were: Age, Gender, Race, Imaging Modality (Echo, CMR, RN), Preoperative EF, History of MI, History of diabetes, History of hypertension, Body mass index, NYHA class III/IV, Creatinine, Beta-blocker, ACE-I or ARB, Aspirin, Clopidogrel, Digoxin, Diuretic loopK sparing diuretic, Nitrate, Pulse, Systolic BP, Number of distal anastomoses, Number of diseased vessels, Total bypass time, Aortic cross-clamp time, CABG + SVR, Current Smoker, Peripheral Vascular Disease, Antiarrhythmic, Baseline BUN, Angina Symptoms, Unstable Angina Symptoms, Ischemic Symptoms, Angina During Exercise, Distance Walked, Myocardial Viability, Postoperative EF, Hyperlipidemia, NHYA class I/II, 6 Minute Walk Test, Duke CAD Index, Statin, Warfarin, ICD Implantation, Death after Randomization, Time to Death

Then, those with a p value <0.1 were entered into the multivariable model. These initial battery of variables were: Preoperative EF, SVR, Digoxin, Warfarin, Hypertension, NYHA III/IV, RN (vs. Echo), and CMR (vs. Echo)(Supplemental Table 1. below). 

Then, backward elimination method was used to eliminate variables with p>0.05 and obtain the most parsimonious model. The final battery of the selected variables were Preoperative LVEF and SVR, which are shown in Table 2 in the manuscript. We have described this methodology in the statistical section of the manuscript (page 6, lines 135-138). The table showing the initial battery of variables was included in the manuscript as a supplemental table because in this pre-backward elimination model the odds ratios would be different than the odds ratios in the final model. 

Text changes (highlighted): “We utilized logistic regression analysis to examine the predictors of EF improvement. All variables that had a p-value ≤0.10 in univariable analysis were entered into a multivariable logistic regression model (Supplemental Table 1). Utilizing backwards elimination, and a more restrictive p-value of ≤0.05, we reached the final multivariable model (16).”

Supplemental Table 1. Odds Ratios of Battery of Potential Predictors for LVEF Improvement

 Variable Odds Ratio 95% CI p-value

Preoperative EF 1.3136 1.17-1.48 <0.001 

SVR 1.6365 1.09-2.45 0.017 

Digoxin 0.5885 0.32-1.08 0.09 

Warfarin 1.559 0.86-2.82 0.14 

Hypertension 0.8477 0.57-1.27 0.42 

NYHA III/IV 1.4322 0.95-2.15 0.08 

RN (vs. Echo) 1.5017 0.85-2.65 0.16 

CMR (vs. Echo) 1.4224 0.91-2.20 0.13 

Abbreviations: CMR=cardiac magnetic resonance, Echo=transthoracic echocardiogram, EF=ejection fraction, NYHA=New York Heart Association, RN=Radionuclide, SVR=surgical ventricular restoration

2. The analysis of perioperative change in LVEF and survival has been adjusted with respect to age and gender. What about all the other predictors displayed in Table 1? I'd welcome a Cox regression model where the predictors were selected by backward elimination, similarly to the approach pursued in the logistic regression analysis. A table with the results of the final Cox model should be then added and discussed. An additional specific point: Fig. 4 displays cumulative incidence curves. Why didn't the authors provide traditional survival curves?

Response: We have updated this analysis by adjusting for additional covariates that were significantly different between the patients with improved vs. unimproved LVEF. These additional variables were baseline LVEF and NYHA Class. As a result, the HR for EF improvement changed from 0.58 to 0.43 (95% CI: 0.26-0.71). We have revised the manuscript accordingly (page 6, lines 142-143; page 10, lines 214-219; page 12 255-256). 

However, backward elimination to identify a multivariable model associated with mortality is beyond the scope of our study. As we stated in the introduction section, the objective of this study was to “characterize the extent, direction and implications of LVEF change following CABG”. We would like to stay focused on that objective. However, if the reviewer and the editors feel strongly about an additional analysis on the predictors of mortality after CABG, we will have to comply. 

With regards to Figure 4, we prefer Cumulative survival curves, which is a mirror image of 1-KM curves shown below. This is a format that we and others have used in many publications. However, if the reviewer and the editors feel strongly that we display the survival curves in the 1-KM format, we will comply. 

Text changes (highlighted): Page 6: “Survival analysis was adjusted for all covariates which were significantly different between patients with vs. without improved LVEF.” 

Page 10: “Over a mean 3.7 (±1.2) years of follow-up, 21/145 (14.5%) vs. 93/404 (23.0%) of patients with or without a �LVEF >10% died, respectively. This translated to a significantly lower risk of all-cause mortality in patients with a perioperative LVEF improvement >10% compared to those with unchanged or decreased LVEF (p=0.027) (Fig 4). After adjusting for age, sex, baseline LVEF, and NYHA Class, perioperative LVEF improvement >10% was associated with a 57% lower risk of all-cause mortality (HR: 0.43, 95% CI: 0.26-0.71) compared to those with unchanged or decreased LVEF.”

Page 12: “In this analysis, we found that a perioperative LVEF improvement >10% afforded a 57% reduction in all-cause mortality after CABG.”

3. The competing risks analysis doesn't seem to be adjusted for the available predictors. Why? I'd welcome an analysis that adjusts for the available predictors, consistently to what has been done in the rest of the paper.

Response: There were 36 deaths due to heart failure and 33 deaths due to SCD, limiting the number of covariates allowable in these models. The limited power of these analyses, evident by the wide 95% confidence intervals, is the reason we could not adjust for additional covariates. We acknowledged the reduced power of these analyses within the manuscript (page 10, lines 225-226; page 12, lines 273-274). Furthermore, since the univariable analysis did not find a significant association, it would be highly unlikely for the multivariable analysis to show one. Despite the low power, we are in the opinion that this analysis adds more depth to the results. However, if the reviewer and the editors want it to be removed, we will comply. 

Text changes (highlighted): “In a competing risk analysis, perioperative LVEF improvement >10% was not associated with the risk of heart failure death (HR: 0.78, 95% CI 0.35 – 1.71; p = 0.53) or sudden cardiac death (SCD) (HR: 0.62, 95% CI 0.26–1.50; p = 0.29), though the statistical power of these analyses was low (36 and 33 total deaths, respectively).”

Reviewer #3:

1) This study is a post-hoc analysis of the STICH trial (2º hypothesis). Although it is implied in the manuscript that this research is a post-hoc analysis of a portion of the STICH trial, this is not clearly stated in the method section of the manuscript.

Response: We have updated the methods section to state the post-hoc nature of the study as suggested (page 5, line 105). 

Text (highlighted changes): “The final cohort for this post-hoc analysis included 549 patients who underwent CABG (+/- SVR), had pre- and postoperative LVEF assessment via identical imaging modalities with good-excellent image quality, evaluated at a STICH core laboratory.”

2) Inclusion criteria are not clearly stated in the manuscript.

Response: We thank the reviewer for this suggestion. We have added the inclusion criteria of the STICH trial in the methods section and provided a reference to the original paper for readers.1 (page 4, lines 93-94)

Text (highlighted changes): “The STICH trial had 2 hypotheses and included 2,136 patients with LV systolic dysfunction, and coronary artery disease amendable to CABG (13).

3) Another question that must be answered is why the patients participating in the first hypothesis of were not included.

Response: Only the patients in hypothesis 2 had successful systematic collection of pre- and post-operative assessment of LVEF at 4 months after CABG. Furthermore, due to the nature of the STICH trial design, patients enrolled in the hypothesis 1 arm were systematically different in baseline ischemic burden than those enrolled in the hypothesis 2 arm. Specifically, to be eligible for enrollment in hypothesis 1 arm, patients could not have either: left main coronary artery stenosis of ≥50% nor a high burden of symptomatic angina (Canadian Cardiovascular Society score III or IV).2 Either, or both, characteristics present on trial enrollment automatically placed the patient in hypothesis arm 2. Given these differences we performed the analysis among patients who participated in Hypothesis 2 of the STICH trial.

4) The study used different methods to evaluate LVEF before and after the procedure. The variations observed in LVEF were comparable among the different methods?

Response: We would like to emphasize that although the study used three different imaging modalities, the present analysis only included patients who were evaluated by the same imaging modality pre- and post-operatively. For example, patients evaluated by echocardiogram preoperatively were only included in the present analysis if the postoperative imaging modality was also echocardiogram. Otherwise, they were excluded to prevent variation in EF stemming from different imaging modalities. 

However, in this comment, the reviewer appears to be asking for the differences between pre- and post-operative EF within each imaging modality. These results are shown in the Table below. We have now included this table as supplemental material. 

Supplemental table 2. Pre- and post-CABG LVEF by imaging modality 

 Preoperative LVEF^ Postoperative LVEF^ Difference in LVEF^ P value

Echo (n = 191) 29.7 (8.1) 33.1 (10.1) 3.4 (10.3) <0.0001*

CMR (n = 273) 27.4 (10.7) 32.6 (13.4) 5.2 (12.1) <0.0001*

RN (n = 85) 27.9 (7.7) 33.8 (9.9) 6.0 (8.1) <0.0001*

Abbreviations: CABG=coronary artery bypass graft, CMR=cardiac magnetic resonance, Echo=transthoracic echocardiogram, LVEF=left ventricular ejection fraction, RN=Radionuclide

^Mean (SD)

*Paired t-test

5) There is no mention in the manuscript how the authors defined the sample size or reason for not having it defined.

Response: As this study was a post-hoc analysis of the STICH trial data, the sample size was determined by the number of patients in the trial that met the inclusion criteria of this post-hoc analysis. Variations in the sample size was not an option due to the retrospective nature of the analysis.

6) SVR did not promote improvement in the original stich trial. How come that it was an independent predictor for > 10 % increase in LVEF? Selection Bias?

Response: To clarify, in the original STICH trial SVR was not associated with mortality (not LVEF). Analysis of pre- and post-operative LVEF was not performed in the original reports from the trial. That is why we felt compelled to perform this analysis.

References (used in responses only, not inclusive of manuscript excerpts)

1. Velazquez EJ, Lee KL, Deja MA, et al. Coronary-Artery Bypass Surgery in Patients with Left Ventricular Dysfunction. N Engl J Med. 2011;364(17):1607-1616. doi:10.1056/NEJMoa1100356

2. Jones RH, Velazquez EJ, Michler RE, et al. Coronary bypass surgery with or without surgical ventricular reconstruction. N Engl J Med. 2009;360(17):1705-1717. doi:10.1056/NEJMoa0900559

---

## [Decision Letter · Decision Letter 1]

28 Oct 2022

Perioperative changes in left ventricular systolic function following surgical revascularization

PONE-D-22-19400R1

Dear Dr. Downey,

We’re pleased to inform you that your manuscript has been judged scientifically suitable for publication and will be formally accepted for publication once it meets all outstanding technical requirements.

Kind regards,

Yoshihiro Fukumoto

Academic Editor

PLOS ONE

Additional Editor Comments (optional):

Reviewers' comments:

Reviewer's Responses to Questions

**Comments to the Author**

1. If the authors have adequately addressed your comments raised in a previous round of review and you feel that this manuscript is now acceptable for publication, you may indicate that here to bypass the “Comments to the Author” section, enter your conflict of interest statement in the “Confidential to Editor” section, and submit your "Accept" recommendation.

Reviewer #1: All comments have been addressed

Reviewer #2: All comments have been addressed

2. Is the manuscript technically sound, and do the data support the conclusions?

Reviewer #1: Yes

Reviewer #2: (No Response)

3. Has the statistical analysis been performed appropriately and rigorously? 

Reviewer #1: Yes

Reviewer #2: (No Response)

4. Have the authors made all data underlying the findings in their manuscript fully available?

Reviewer #1: Yes

Reviewer #2: (No Response)

5. Is the manuscript presented in an intelligible fashion and written in standard English?

Reviewer #1: Yes

Reviewer #2: (No Response)

6. Review Comments to the Author

Reviewer #1: The authors examined changes in LVEF in 549 patients with left ventricular systolic dysfunction (LVEF <50%) who underwent CABG as part of the Surgical Treatment of Ischemic Heart Failure (STICH) trial.

In conclusion, approximately 25% of patients with ischemic cardiomyopathy undergoing CABG experienced perioperative LVEF improvement of 10% or more, the likelihood of LVEF improvement was inversely proportional to preoperative LVEF, LVEF improvement was not affected by the presence of myocardial viability, and LVEF improvement was associated with improved long-term survival. These results add to our understanding of perioperative LVEF changes related to CABG by demonstrating an inverse correlation between LVEF improvement and preoperative LVEF.

The manuscript was improved as a result of peer review. No further comments.

Reviewer #2: (No Response)

7. PLOS authors have the option to publish the peer review history of their article (what does this mean?). If published, this will include your full peer review and any attached files.

Reviewer #1: No

Reviewer #2: No
